# An Optical Coherence Tomography-Based Measure as an Independent Estimate of Retinal Function in Retinitis Pigmentosa

**DOI:** 10.3390/diagnostics13233521

**Published:** 2023-11-24

**Authors:** Manuel Paez-Escamilla, Michelle L. Alabek, Oliver Beale, Colin J. Prensky, Raphael Lejoyeux, Thomas R. Friberg, Jose-Alain Sahel, Boris Rosin

**Affiliations:** 1Department of Ophthalmology/UPMC Vision Institute, University of Pittsburgh Medical Center (UPMC), 1622 Locust Street, Pittsburgh, PA 15219, USA; paezin@gmail.com (M.P.-E.); alabekm@upmc.edu (M.L.A.); bealeo2@upmc.edu (O.B.); prenskyc@upmc.edu (C.J.P.); lejoyeux.raphael@gmail.com (R.L.); friberg@pitt.edu (T.R.F.); sahelja@upmc.edu (J.-A.S.); 2Department of Ophthalmology, McGill University Health Centre, Montreal, QC H4A 3J1, Canada; 3Rothschild Foundation Hospital, 75019 Paris, France; 4Institut Oeil Paupiere, Viry-Chatillon, 91170 Paris, France

**Keywords:** retinal reflectivity, retinal dystrophies, retinitis pigmentosa, visual acuity, retinal function, optical coherence tomography, reflectivity ratio

## Abstract

Background: With the clinical advances in the field of gene therapy, the development of objective measures of visual function of patients with inherited retinal dystrophies (IRDs) is of utmost importance. Here, we propose one such measure. Methods: We retrospectively analyzed data from a cohort of 194 eyes of 97 genetically diagnosed patients with retinitis pigmentosa (RP), the most common IRD, followed at the UPMC Vision Institute. The analyzed data included the reflectivity ratio (RR) of the retinal nerve fiber layer (RNFL) to that of the entire retina, visual acuity (VA) and the thickness of the retinal outer nuclear layer (ONL) and the RNFL. Results: There was a strong positive correlation between the RR and VA. Both VA and the RR were negatively correlated with disease duration; VA, but not the RR, was negatively correlated with age. The RR correlated with the ONL but not with the RNFL thickness or the intraocular pressure. Age, RR, disease duration and ONL thickness were found to be independent predictors of VA by multivariate analysis. Conclusion: The OCT RR could serve as an independent predictor of visual acuity, and by extension of retinal function, in genetically diagnosed RP patients. Such objective measures can be of great value in patient selection for therapeutic trials.

## 1. Introduction

IRDs, which comprise a highly heterogeneous group of visual disorders [1], usually result in profound visual impairment in patients. With an incidence of 1 in 3000, these disorders result in a substantial socioeconomic burden due to the high cost of medical care and decreases productivity in the workforce [2]. Furthermore, the psychological burden of these disorders on patients is profound [3]. Among all IRDs, retinitis pigmentosa (RP) is the most common condition, affecting approximately 1 in 4000 individuals [4]. RP exhibits significant genetic heterogeneity, with many genetic defects resulting in the same phenotype [4]. Such heterogeneity introduces incredible complexity to various gene therapy approaches, requiring standardization of the outcome measures of any interventional therapy [5]. With gene therapy for *RPE65*-associated retinal dystrophy being clinically available [6] and gene therapy for many other IRDs being developed [7], it is paramount to develop objective non-invasive measures of visual acuity and retinal function that can be used both to aid in patient selection for treatment and clinical trials as well as in the estimation of treatment effectivity. Here, we identify a measure that uses the optical coherence tomography (OCT) reflectivity property as defined below. Naturally, additional OCT, and in particular OCT reflectivity-based methods, should continue to be developed.

To better elucidate the motivation for this study, first, we provide an intuitive explanation of, and later define, the concept of reflectivity and relative reflectivity. When comparing the OCT scans of healthy subjects (Figure 1D, upper panel) to those of RP patients (Figure 1A–C, upper panels), two important differences stand out. First, there are obvious changes in the volume of the different retinal layers and of the entire retina, which have been previously described by other groups [8]. However, when comparing the individual retinal layers, it becomes clear that another apparent difference exists in addition to the volumetric changes. This is the difference in the “contrast” or “intensity” of the different layers as depicted by the OCT imaging. This variability may also be seen as an increase or decrease in apparent contrast between retina layers within an individual scan. For instance, the retinal nerve fiber layer (RNFL) of the control subject is significantly brighter than their outer nuclear layer (ONL) (Figure 1D, upper panel), whereas the differences in intensity between the RNFL and ONL within RP patients (Figure 1A–C, upper panels) are not as profound. This is the property we set out to quantify and compare between patients and controls, using an imaging parameter wedged at the core of OCT imaging, i.e., tissue reflectivity.

OCT, which is a widely utilized technique for the imaging of ocular structures, allows the visualization of retinal layers at a microscopic level [9]. OCT utilizes light waves to render an image of retinal structure in a manner similar to the use of sound waves in ultrasonographic studies. The basic units of information of OCT are unidimensional A-scans, which are measures of retinal tissue light reflectivity at specific depths of the retina. These are subsequently combined into two-dimensional B-scans that provide high-resolution images of the retinal layers, which are widely used for analysis of retinal structure in health and disease [10]. Thus, retinal tissue reflectivity is the core parameter measured by OCT and it is expressed in the B-scans by the brightness of the individual pixels comprising the image.

Since its introduction, retinal OCT has advanced to offer microscopic resolution of the retinal structures [11]. The introduction of swept-source OCT has allowed greater sensitivity of the imaging modality and has improved the signal-to-noise ratio at greater tissue depths in both animal models of retinal disorders and human patients [12,13]. Furthermore, the use of sequential high-resolution images of the OCT has introduced a novel angiography modality through interscan comparison, OCT angiography [14]. In addition, small OCT interscan changes have been demonstrated to occur in the initial phase of phototransduction and have been found to be repeatable and quantifiable. This, in turn, introduced novel functional test bridging imaging and psychophysics, termed optoretinography [15]. It is safe to say that OCT imaging has become the mainstay of any retinal practice, allowing continuous follow-up, objective estimation of retinal status and clinical decision making in various pathologies [16,17,18,19]. Its importance only grew during the COVID-19 pandemic, where remote evaluation of OCT imaging allowed decision making in a challenging clinical environment [20].

Specifically, the analyses of changes in the reflectivity of different retinal findings have yielded interesting and highly useful results. Hypo-reflective changes have long been used for the sequential quantification of intraretinal and subretinal fluid accumulation in various retinal disorders, recently employing deep learning algorithms [21]. Such changes have also been shown to accompany neovascular complexes in age-related macular degeneration (AMD) and diabetic retinopathy [22,23] and have been demonstrated in other less common pathologies [24].

Hyper-reflective changes have been shown to exist in various clinical and surgical retinal pathologies, and their existence has been shown to be of clinical importance. Examples include hyper-reflective lines that are predictive of macular hole development following a vitrectomy [25,26] as well as focal hyper-reflective changes predictive of a poorer prognosis in diabetic retinopathy with diabetic macular edema (DME) [27].

In addition to the descriptive reports of hyper-reflective changes, objective quantification of such changes has been shown to be useful, albeit challenging. Use of this approach in patients with pseudoxanthoma elasticum identified that the hyper-reflectivity of the Bruch’s membrane represented its progressive calcification, heralding visual morbidity [28]. Additionally, in patients with DME, an algorithm for quantification of the previously mentioned intraretinal hyper-reflectivity has recently been described [29].

Therefore, we sought to analyze the OCT scans of RP patients and healthy controls, focusing on the reflectivity properties of the OCT scans. With the inner retina coming into focus as a potential target for several therapeutic interventions in IRDs, we have focused our efforts on inner retinal structures, particularly the RNFL. In a recent study, optogenetic gene therapy targeted at the inner retina has been shown to be effective in restoring vision in blind subjects [30]. Naturally, both optogenetics as well as retinal prostheses, another novel restorative approach, require a functioning RNFL in order to relay the visual information to the central nervous system [31]. With these caveats in mind, we have decided to focus our study on the inner retina and specifically on the RNFL, as the final output layer of the retina.

## 2. Materials and Methods

We performed a retrospective analysis of patients with RP and positive genotyping from a tertiary ophthalmology center with a dedicated IRD service. The analyzed data included visual acuity (VA, ETDRS scale), intraocular pressure (IOP, mmHg), optical coherence tomography (OCT) imaging and genetic test results. The thickness of the ONL and RNFL (µm) were measured by two members of the study team to ensure reproducibility. Genotyping with a single pathogenic or likely pathogenic variant in a dominant gene associated with RP or a combination of at least 2 pathogenic or likely pathogenic variants in a recessive gene associated with RP was considered positive. We designed a custom algorithm to measure the reflectivity ratio (RR) and the mean reflectance of two areas of an OCT image in MATLAB (2023a, Mathworks, Natick, MA, USA). The mean reflectance for a given layer was computed as the mean pixel intensity of the user-delineated region. For each scan, the RR was manually delineated. For a subset of images (*n* = 15), this analysis was performed by 2 different observers in a masked manner, demonstrating high level inter-observer agreement employing predefined protocols regarding distance of measurements from the fovea. When comparing measurements from two observers, they exhibited a highly significant correlation coefficient (r = 0.98, *p* = 0.143 × 10^−13^). We employed this algorithm to quantify the relative reflectance of the retinal nerve fiber layer (RNFL) to the entire retina (Figure 1). Use of ratios between the reflectivity of layers for each scan instead of comparing directly between the intensity values for different images allowed controlling for interscan acquisition parameter variability and performing statistical analysis. A correlation analysis was used to estimate the inter-relations between the various clinical and imaging parameters. A linear regression analysis was chosen in Plots 2–4 as a means of graphic representation of the sign of the correlation coefficients for the different parameters. A multivariate analysis was used to ascertain the independence of the effect of each examined parameter on the visual function, expressed by the visual acuity of our patient cohort. All statistical analyses were performed in MATLAB.

## 3. Results

### Analyses

The cohort included 97 patients with RP spanning 39 different genetic diagnoses, with *USH2A* presenting as the most common causative gene (22/97r, 22%). The RR value for each eye of each patient (Figure 1) was comparable between the eyes (OD 1.3474 ± 0.0024 and OS 1.3397 ± 0.0023, mean ± SEM). Our analysis revealed a strong positive correlation between the RR and VA in both eyes analyzed independently (OD, r = 0.6224, *p* = 7.89 × 10^−12^ and OS, r = 0.5652, *p* = 1.623 × 10^−9^ (Figure 2)). Age was negatively correlated with VA in both eyes (OD, r = −0.3178, *p* = 0.0245 and OS, r = −0.4518, *p* = 0.001). In line with previous results, the duration of disease from diagnosis was also negatively correlated with VA (OD, r = −0.517, *p* = 4.99 × 10^−8^ and OS, r = −0.56, *p* = 2.05 × 10^−9^ (Figure 3)). There was a weak trend towards a negative correlation between age and the OCT reflectivity ratio for both eyes (OD. r = −0.161, *p* = 0.1146 and OS, r = −0.1654, *p* = 0.1055) which failed to reach significance. There was a significant negative correlation between the duration of disease and the OCT reflectivity ratio for both eyes (OD, r = −0.319, *p* = 0.0013 and OS: r = −0.272, *p* = 0.0065). Multiple regression analysis revealed both age and the OCT RR as independent predictors of visual acuity for each eye (OD, *p* = 3.87 × 10^−14^ and OS, *p* = 0.1.88 × 10^−14^). By virtue of the RR being a ratio, larger values can represent an increased numerator, a decreased denominator or both. The ONL is the main contributor to retinal hypo-reflectivity, ultimately affecting the denominator of the RR, while the RNFL is the only layer affecting the numerator of the RR. Therefore, we computed the maximal ONL (OD 53.2887 µm ± 0.3554 and OS 58.2577 µm ± 0.5835) and RNFL (OD 44.2577 µm ± 0.1757 and OS 40.5258 µm ± 0.1414) thicknesses for each retinal scan. While there a significant correlation between the RR and ONL thickness in each eye (OD, r = 0.408, *p* = 3.337e-05 and OS, r = 0.213, *p* = 0.0364), there was no significant correlation between the RR and RNFL thickness in either eye (OD, r = 0.188, *p* = 0.043 and OS, r = 0.1574, *p* = 0.124). When examining the correlation coefficient between the RR and the intraocular pressure (IOP) for each eye, there was a trend towards a negative correlation, failing to reach statistical significance (OD, r = −0.1069, *p* = −0.33 and OS, r = −0.2432, *p* = 0.16). As expected, the ONL thickness was positively correlated with VA for each eye (OD, r = 0.653, *p* = 3.84 × 10^−13^ and OS, r = 0.609, *p* = 4.597 × 10^−13^ (Figure 4)). There was no correlation with VA and RNFL thickness in either eye (OD, r = 0.175, *p* = 0.087 and OS, r = −0.144, *p* = 0.158). Notably, the multiple regression analysis revealed the RR and ONL thickness to be independent predictors of VA for each eye (OD, *p* = 0.039 and OS, *p* = 0.016).

In order to account for the effect of patients with poor vision on the cohort analysis, we performed a correlation analysis excluding the subgroup of patients with NLP vision. As expected, the RR remained significantly correlated with VA for each eye (OD, r = 0.39, *p* = 1.93 × 10^−5^ and OS, r = 0.302 and *p* = 1.93 × 10^−5^). Interestingly, these correlation values are more moderate than the values received when including the blind patients (OD, r = 0.6224, *p* = 7.89 × 10^−12^ and OS, r = 0.5652, *p* = 1.623 × 10^−9^, Figure 2). This implies that adding the patients with NLP vision strengthens the correlation between the RR and VA, indicating that patients with NLP vision tended to have lower values of RR as compared with seeing patients.

Further analysis between genotypes was not performed due to sample size limitations.

While the intent of this work was to focus on analyzing the RR in RP patients with a confirmed genotype, the analysis was also performed for eight eyes of eight healthy-sighted controls. The relative ratio in this cohort was 1.5582 ± 0.0160 (mean ± SEM (Figure 1D)). Even for such a small cohort, the difference between the RR values among controls and RP patients was significant when comparing to both eyes of RP patients (*p* = 0.012 for OD, *p* = 0.0145 for OS, two-sample *t*-test), reinforcing the validity of the RR as an estimate of visual acuity and, by extension, retinal function.

## 4. Discussion

Here, we devise a new OCT metric based on relative reflectance of the RNFL to the entire retina. While not fully understood, the reflectance of the different retinal layers on OCT imaging is thought to represent intrinsic tissue properties [32]. Repeatability of OCT scans within and between visits has been a concern for volumetric measurements [33] as well as for OCT angiography, where a comparison of interscan intensity has yields the resultant blood flow [34]. While comparing the pixel intensity between different scans can be problematic due to differences in acquisition parameters, we overcame this difficulty by using ratios of reflectivity within the same scan, which, assuming uniform acquisition parameters variation within an individual scan, allowed for interscan comparisons.

In this study, we focused on the RNFL for several reasons. The requirement of a functional RNFL is pivotal for the application of the aforementioned optogenetics [30] and retinal prostheses [31]. Since the RNFL is the output stage of the visual pathway [35], the existence of a functioning RNFL is a prerequisite to vision restoration employing essentially any treatment method for any retinal condition targeting the retina itself, including IRDs. The only conceivable exception to this rule is the field of multielectrode stimulation of the CNS, bypassing the retina and the optic nerve [36]. As such, it makes the study of the RNFL of utmost importance when selecting patients for trials and therapies targeting the retina. Interestingly, the RNFL volumetric changes during the course of RP progression are not straightforward, with some researchers reporting gene-specific thickening of the RNFL over the course of the disease [37,38], while others have even reported gene-independent thickening of the RNFL [39]. Thus, OCT measurements of the RNFL can provide robust and ample data even in the late stages of disease, when the other retinal layers have undergone significant atrophic changes [39].

Notably, the term “gene therapy” encompasses much more than mere replacement therapy, the approach used for *RPE65*-associated retinal dystrophy. Optogenetics, or the introduction of light-sensitive opsins into neurons, was introduced almost two decades ago [40,41]. This approach is particularly interesting for advanced stages of IRDs, where it has been shown that the inner retinal cells tended to survive much longer than the photoreceptors [42]. As mentioned, a recent study demonstrated restoration of visual function in blind subjects after the introduction of an algae-derived opsin ChrimsonR into the surviving inner retinal cells [30]. Another gene therapy-based approach is the introduction of trophic factors, such as the rod-derived cone viability factor (RdCVF), into the retina to promote the survival of existing photoreceptors [43]. All such approaches require a mass of surviving retinal cells to express the opsin or the trophic factor in place and allow the added function of the inner retinal cells or the rescue of the remaining photoreceptors [4]. Finally, retinal prostheses, several types of which exist [44], have already been implanted in human patients of AMD [31]. They also require an intact inner retina and specifically RGC and RNFL to relay the visual information from the prosthetic device to the CNS [45]. Studies of the interrelation of photoreceptor and inner retina well-being are ongoing [46]. For all of the above, we felt that the focus on the inner retina of this study was appropriate.

As stated above, by virtue of the RR being a ratio, larger values can represent an increased numerator, a decreased denominator or both. Since the thickness of the measured retinal layers is fully expected to affect the computed average intensities, volumetric changes, undoubtedly, affect the computed values of the RR. However, this effect is by no means straightforward. Reinforcing the intuitive assumptions about dystrophies, it is true that studies of OCTs in RP patients have shown progressive thinning of the retinal thickness [8]. However, and quite unexpectedly, this does not hold true for all retinal layers in all subtypes of IRDs. Such is the case, for instance, for *RPGR*-related RP and *ABCA4*-related diseases, where counter-intuitive thickening of some layers has been demonstrated [37,38]. To reinforce this point, in our study, while the RR correlated with ONL thickness, it did not correlate with RNFL thickness. Thus, the effect of volumetric changes on the RR is complex, likely differing between various genetic conditions and does not provide a straightforward explanation for the correlation found between the RR and VA in this study.

We believe that the RR is strongly affected by the intrinsic properties of the imaged tissues and not mere changes in volumes of the retinal layers. There is ample evidence to support this hypothesis. For instance, waxy pallor of the optic nerve head, a classic clinical finding in advanced RP, has been shown to correlate with the presence of a glial membrane over the optic nerve head, which if extending over the RNFL will undoubtedly affect its intensity [47]. Furthermore, the effect of lutein supplementation on macular pigment has been established in age-related macular degeneration [48], choroideremia [49], *ABCA4*-related disease [50] and RP [51]. All these and others have the potential to affect OCT reflectance and, in turn, the RR.

Our results indicate that the RR correlated with the VA of the patients in our cohort. While age, disease duration and ONL thickness also showed significant correlations with VA in this study, there were several important differences. It is of no surprise that age and disease duration are inversely correlated with vision in our cohort, as RP is a progressive disorder [3]. The correlation of the ONL thickness, indicative of the number of surviving photoreceptors, and VA is also by no means a surprise, as it has been demonstrated in health and retinal disease [52]. However, age, disease duration and ONL thickness have been shown to correlate with VA independently of the RR when utilizing a multivariate analysis. Thus, parameters based on the OCT reflectivity properties are independent of other well established volumetric and epidemiologic predictors of VA in RP. As mentioned before, the RNFL thickness did not prove to be significantly correlated with the RR, which, taken together with the multivariate analysis results, reinforces the concept that the RR’s value is not a simple derivative of volumetric changes. The RR was also not significantly correlated with the IOP, ruling out glaucomatous changes in RP patients as the source for variation in RR. To the best of our knowledge, this is the first work correlating parameters of OCT reflectivity and visual function in RP patients. As such, it is of utmost importance when approaching experimental trials and estimating treatment outcomes.

Visual acuity is usually poor in advanced IRDs, as is the case in Leber’s congenital amaurosis, the classic *RPE65*-related disease, where poor visual acuity from birth was one of the defining features of the disease described by Leber in 1869 [53]. Alternative methods for the estimation of visual acuity are often used in children and non-verbal adults. In these populations, both visual acuity and contrast sensitivity could be estimated by objective measures employing vision-dependent reflexes, such as the optokinetic nystagmus (OKN) [54], although the relationship between these parameters is not straightforward [55]. Such is also the case for laboratory animals, where OKN testing is the paradigm of choice for VA estimation [56]. However, the implications of the results of this study to laboratory animals are not immediate. For instance, rodent models of IRDs, the main pool of animal models for these disorders, lack a macula, a fact that will undoubtedly affect the computed values of the RR [57]. In human patients, other approaches such as patient-reported questionnaires have also been employed [58]. The multitude of approaches is indicative of a real need for methods of fast and non-invasive estimation of visual acuity, making the strong correlation between the RR and VA in our study of particular interest. Interestingly, when analyzing the RR in our cohort excluding the patients with NLP vision, a less significant correlation was observed between the RR and VA as compared to the entire cohort. This implies that patients with NLP vision tended to have lower values of RR as compared with seeing patients. Notably, while we chose linear regression to simplify the graphic representation of the sign of correlations between the parameters depicted in Figure 2, Figure 3 and Figure 4 this relation is by no means linear. While visual acuity and visual function are correlated, they are not identical. Visual function, defined as the useful processing of visual information, is of particular interest in IRDs and has been shown to be restored through novel treatment modalities in patient exhibiting very poor and stable visual acuity before and after treatment [30]. We did not explore the relationship of RR to visual function, and future studies are required in order to establish such a relationship.

This study has certain limitations. This is a retrospective study, and as such, it lacks randomization as well as a longitudinal follow-up component. The number of patients is limited by the cohort followed at our Center, and larger numbers could shed more light on the question of validity of the RR. Future studies are needed to examine the relative reflectance of additional retinal layers and their correlation to visual function parameters. Ideally, the latter should include not only visual acuity, but also other parameters such as electroretinography responses. Finally, while the analysis in this manuscript was performed manually, one can envision such OCT-based parameters being incorporated in artificial intelligence algorithms aimed at determining the functional status of IRD patients. Nevertheless, this study can serve as a proof of concept for objective OCT-based measures in the estimation of retinal function in patients with IRDs.

To summarize, as we enter a new era of gene therapy and prosthetics being readily available for IRDs, the ability to objectively quantify the visual acuity and the visual function of potential candidates becomes paramount. Undoubtedly, there would be no single measure of outcome of such therapies, and a multifactorial approach would be required, including imaging-based techniques [59]. With a multitude of new studies and trials being initiated at the present time, methods such as the one introduced in this study could be employed in more than one capacity. Primarily, they could serve as outcome measures, both for observational and interventional studies. However, and perhaps more importantly, they could provide the means of patient selection for such studies, ensuring patient safety and preventing damage in an objective and a non-biased manner. Such approaches could also be useful in determining the patients most likely to benefit from such interventions. This study presents one such approach and undoubtedly additional approaches, including OCT-based approaches, should continue to be developed.

## Figures and Tables

**Figure 1 diagnostics-13-03521-f001:**
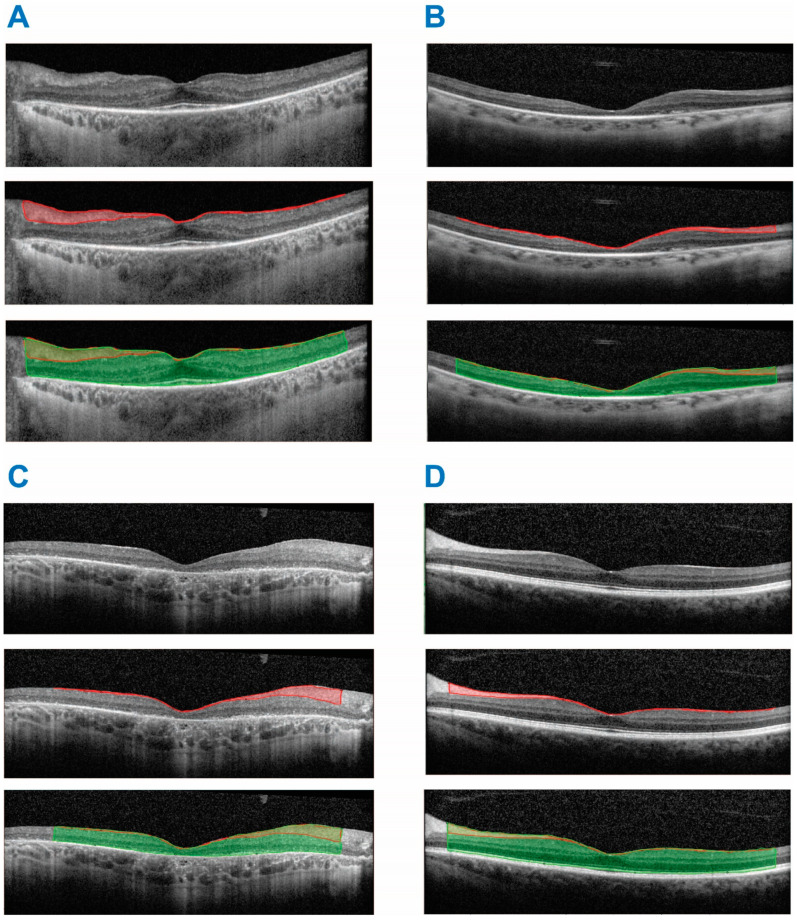
Examples of algorithm calculations for four patients demonstrating a raw OCT scan (top row), OCT with the RNFL marked in red (middle row) and an OCT with the entire retina marked in green (bottom row). The RR is calculated as the ratio between the mean reflectance of the area delineated in the middle row in red (RNFL) divided by the mean reflectance of the area delineated in the bottom row in green (the entire retina). Examples include RP patients in (**A**–**C**) with (**A**) *RHO* mutation, RR = 1.16; (**B**) *RPGR* mutation, RR = 1.32; (**C**) *USH2A* mutation, RR = 1.27; (**D**) normally sighted control, RR = 1.69.

**Figure 2 diagnostics-13-03521-f002:**
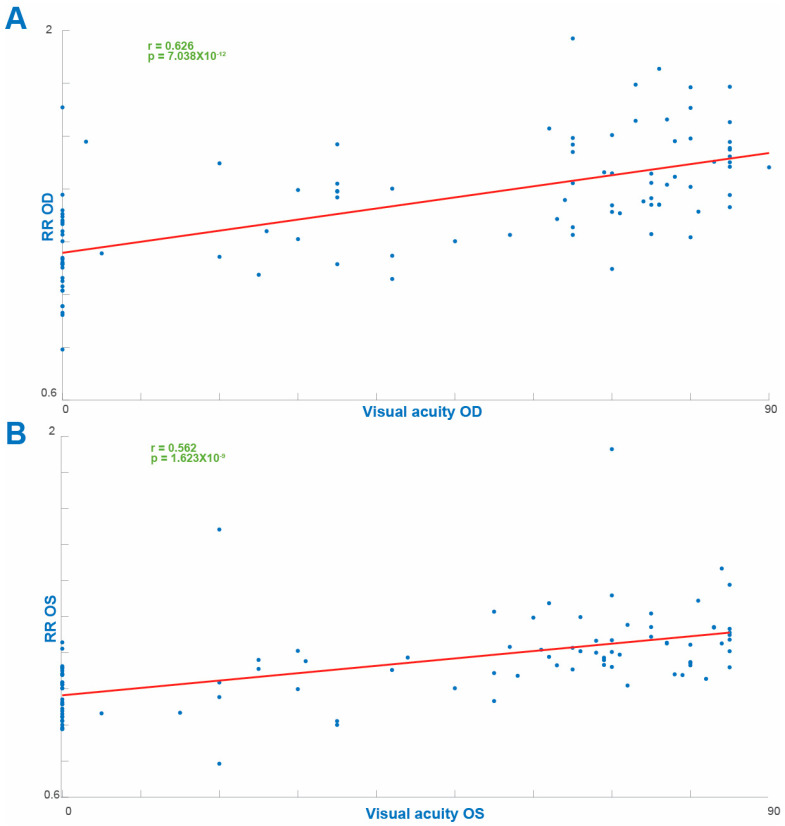
Correlation coefficients between the reflectivity ratio (RR) and visual acuity for the right ((**A**), upper panel) and the left ((**B**), lower panel) eyes.

**Figure 3 diagnostics-13-03521-f003:**
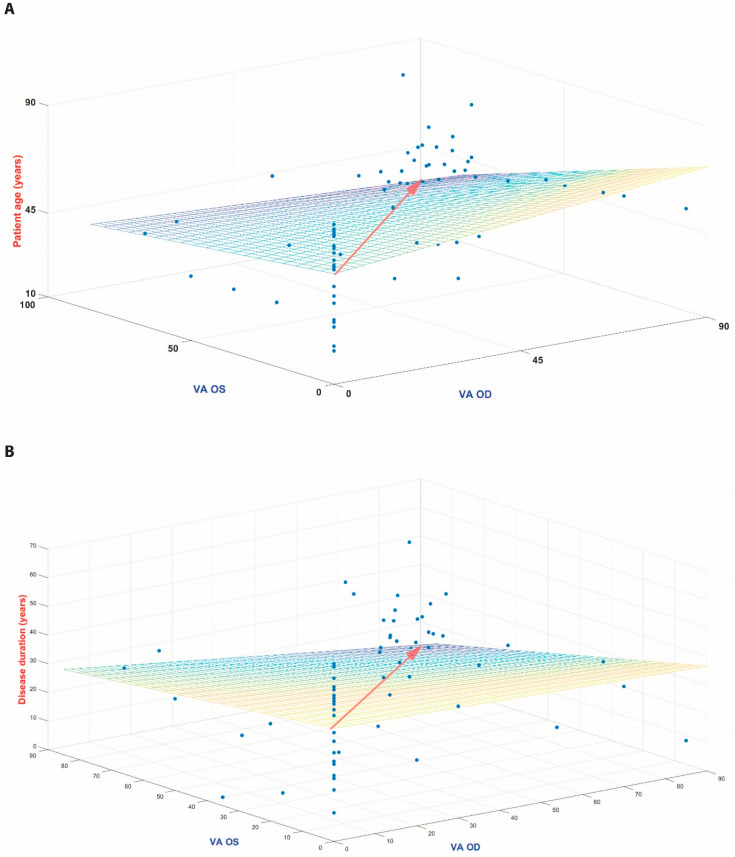
Three-dimensional representation of the negative correlation of both age (**A**) and disease duration (**B**) with visual acuity for each eye. The main diagonal (transparent arrow) of the approximated plane through the scatterplot of age as a function of visual acuity for OD (*x*-axis) and OS (*y*-axis) demonstrates a downwards slope (higher VA for younger age and shorter disease duration for (**A**) and (**B**), respectively).

**Figure 4 diagnostics-13-03521-f004:**
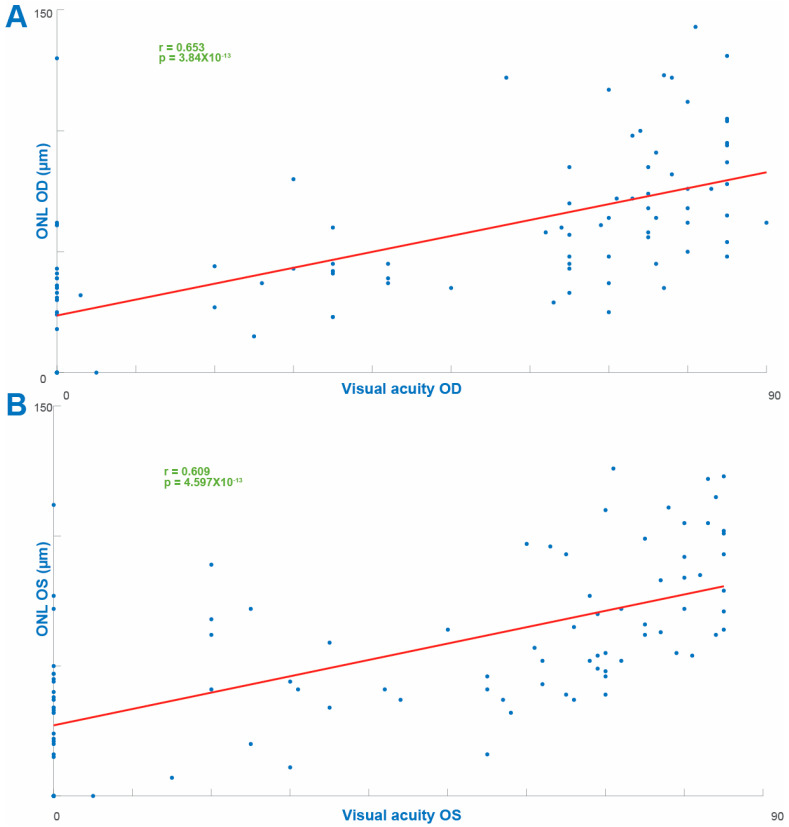
Correlation coefficients between visual acuity and ONL thickness for the right ((**A**), upper panel) and the left ((**B**), lower panel) eyes.

## Data Availability

The data that support the findings of this study can be made available upon request from the corresponding author, B.R., under certain conditions. The data are not publicly available due to their containing information that could compromise the privacy of research participants. Release of clinical data is subject to UPMC regulations and approval.

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
