# Peer review of "An Optical Coherence Tomography-Based Measure as an Independent Estimate of Retinal Function in Retinitis Pigmentosa"

_diagnostics, 2023, doi:10.3390/diagnostics13233521_

Round 1

Reviewer 1 Report

Comments and Suggestions for Authors

This work is of great clinical value.

I have several remarks to improve.

First, have you measured intraocular pressure and look for correlation between it and RR in OCT?

Second,  For instance, you stated that the retinal nerve fiber layer (RNFL) of the control subject  is significantly brighter than their outer nuclear layer (ONL; figure 1D, upper panel) whereas the differences in intensity  between the RNFL and ONL within RP patients (figure 1A-C, upper panels) are not as profound. Have you compared same parameters in patients with glaukoma or blind ones? 

Third, I have no information in Discussion part on comparison cell lines or animal models to prove working protocol? Is that automated protocol or manually checked by operator? Is that possible to introduce AI to identify pathological changes in ONL and RPNL?

Author Response

Please find attached our detailed response to reviewers 1 and 2 in an MS-word file. We thank both reviewers for the time taken to evaluate our work and for their constructive comments.

Reviewer 2 Report

Comments and Suggestions for Authors

The paper by Paez-Excamilla and coworkers is very interesting and may provide an important new assessment tool that could be quite useful for clinical trial execution as well as routine clinical care. I ask the authors to address the questions/comments below.

1.     Abstract: “There was a 19 strong positive correlation between the RR and VA, while both were negatively correlated with 20 disease duration and the VA but not the RR was negatively correlated with age.” This sentence is confusing. Please clarify.

2.     Results: “affective” should be “affecting”.

3.     I’m a little confused by the linearity of the graphs in Figures 2-4. If I have understood the calculation correctly, once the patient has no light perception or perhaps only light perception the RR ratio should reach a stable value since the ONL cannot become any thinner and therefore the overall reflectivity (bottom row of photos in Figure 1) should reach a stable value. Furthermore, the hill of vision (relationship between distance from fovea and visual acuity) is nonlinear. So I expect the curves in Figure 1 not to be straight lines as the area sampled is large compared to the area of the fovea, and much of the retina degenerates in RP while the patient maintains VA 20/20 as the fovea is preserved until very late stages of the disease. Was a curve fitting program used to identify best fit for the lines? Or perhaps I have misunderstood the analytical basis of the graphs.

Author Response

Please find attached our detailed response to reviewer 1 and 2 in an MS-word file. We thank both reviewers for the time taken to evaluate our work and for their constructive comments.
